

# Distribution of crabs along a habitat gradient on the Yellow Sea coast after *Spartina alterniflora* invasion

Pan Chen[1,2], Yan Zhang[1], Xiaojing Zhu[1] and Changhu Lu[1]

[1] College of Biology and the Environment, Nanjing Forestry University, Nanjing, Jiangsu, China
[2] College of Environment and Engineering, Anhui Normal University, Wuhu, Anhui, China

## ABSTRACT

The effects of *Spartina alterniflora* invasion on macrobenthos have long been of concern; however, there is currently no unified conclusion regarding these effects. Most studies on crabs focus on one species or limited habitat types, and assessments of the community-level effects of *S. alterniflora* invasion considering multiple species and habitat types have rarely been conducted. In this study, we sampled crabs along a habitat gradient from the shoreline to inland areas on the Yellow Sea coast, including the mudflat, *S. alterniflora* marsh, *Suaeda salsa* marsh and *Phragmites australis* marsh. A total of 10 crab species were found among all habitats, with five species in the mudflat, six species in *S. alterniflora* marsh, seven species in *S. salsa* marsh and four species in *P. australis* marsh. The Shannon index values for the crab communities were similar between *S. alterniflora* marsh and *S. salsa* marsh, and these values were significantly higher than those for the mudflat and *P. australis* marsh. However, the total biomass of crabs was highest in the mudflat, and *Metaplax longipes*, *Philyra pisum* and *Macrophthalmus dilatatus* exclusively preferred the mudflat. The analysis of principal components and similarities showed that the crab community structure in *S. alterniflora* marsh was most similar to that in *S. salsa* marsh, while the crab community structure in the mudflat was most different from that in the other habitat types. Our results demonstrate that the distribution of crabs varies across a habitat gradient after *S. alterniflora* invasion and that the crab community in *S. alterniflora* marsh is slightly different from that associated with the local vegetation but shows a large difference from that in the mudflat. This study indicates that some crab species may have adapted to habitat containing alien *S. alterniflora*, while other crab species reject this new marsh type. The effects of the distribution of crabs after *S. alterniflora* invasion on the regional ecosystem need further study in the future.

Corresponding author
Changhu Lu,
luchanghu@njfu.com.cn

## INTRODUCTION

Biological invasions often threaten the existence of native species and potentially destroy the structure and function of ecosystems, which is one of the essential causes leading to the loss of global biodiversity (*Butchart et al., 2010*; *Genovesi et al., 2015*). Exotic plants

can strongly alter the biotic and abiotic traits of local ecosystems (*Sheehan & Ellison, 2014*), affecting many taxa, such as plants, insects, birds and benthos (*Li et al., 2009*). *Spartina alterniflora* (Poaceae), which is native to the Atlantic and Gulf coasts of North America, is one of the most notable plant invaders, with strong fecundity and ecological adaptability (*Nishijima, Takimoto & Miyashita, 2016*). *S. alterniflora* was deliberately introduced into Chinese tidal flats in 1979 to help with erosion control, dike protection and soil amelioration (*Gao et al., 2012*; *Zuo et al., 2012*). However, *S. alterniflora* often shows strong competitiveness and rapidly replaced native plants, reshaping the landscape pattern in invaded areas (*Yang et al., 2017*). Extensive *S. alterniflora* invasion has caused a series of ecological consequences to native plants, birds and microbenthic invertebrate communities, especially decreased biodiversity and the degradation of native habitats (*Liu et al., 2012*).

The invertebrate macrobenthos is an important biological group in coastal ecosystems and affects sediment stability, controls energy flow and nutrient cycling and influences other taxa (*Levin & Talley, 2002*; *Reise, 2002*). The effects of *S. alterniflora* invasion on macrobenthic communities has attracted much attention, but there is no unified conclusion (*Quan et al., 2016*). The responses of different benthic groups to *S. alterniflora* invasion differ. Based on various and contrasting results, *S. alterniflora* invasion mainly inhibits infauna, such as bivalves, polychaetes and oligochaetes, causing the decline of biomass and biodiversity (*Brusati & Grosholz, 2006*). However, invasion facilitates surface feeders, such as crabs, and some studies have verified that *S. alterniflora* invasion can offer suitable habitat and food for some crab species (*Gao et al., 2018*; *Wang et al., 2008*). Another study showed that the total number and biomass of crabs were much higher in *Spartina*-invaded habitats than in noninvaded habitats; however, the species richness and Shannon diversity were much lower (*Cui, He & An, 2011*). Therefore, the effects of *S. alterniflora* invasion on crabs may be species-specific in terms of different habitat and food requirements (*Wang et al., 2015*). The abovementioned studies mostly focused on one species or limited habitat types, and assessments of the community-level effects of *S. alterniflora* invasion considering multiple species and habitat types have rarely been reported.

In this study, we selected one area of *S. alterniflora* invasion in a Yellow Sea tidal marsh with four habitat types and high crab biodiversity. The effect of *S. alterniflora* invasion on crabs was studied along a habitat gradient from the shoreline to inland areas, including the mudflat, *S. alterniflora* marsh, *Suaeda salsa* marsh and *Phragmites australis* marsh. Our study had two aims: (1) to identify the differences in crab community structure in different habitats after *S. alterniflora* invasion and (2) to identify the temporal and spatial trends shown by crabs according to habitat type. This study may provide more detailed information for future research on the effect of *S. alterniflora* invasion on macrobenthos.

## MATERIALS AND METHODS

### Study site

Field work was conducted in the core area of Yancheng National Nature Reserve (32°59′N–33°03′N, 120°47′E– 120°53′E) in Jiangsu Province, China, from April to
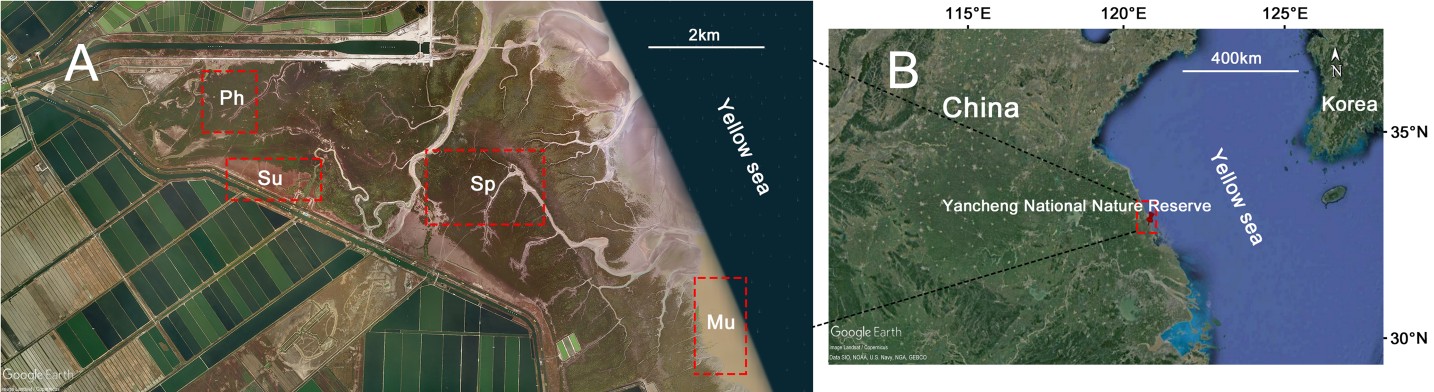

**Figure 1** **Maps of habitat types and sample sites at Yancheng National Nature Reserve.** (A) Gradient distribution of habitat types in the study area. (B) The location of Yancheng National Nature Reserve in China. Habitat types: Mu, mudflat; Sp, *S. alterniflora* marsh; Su, *S. salsa* marsh; Ph, *P. australis* marsh. Map data © 2017 Google.           

September in 2017. The reserve is a typical intertidal wetland located along the Yellow Sea coast in eastern China (two to four m above sea level); it has high biodiversity and serves as an important stopover for a variety of migratory shorebirds (*Melville, Chen & Ma, 2016*). The annual average temperature is 14.1 °C, with the lowest monthly mean temperature of 0.8 °C occurring in January and the highest monthly mean temperature of 27.0 °C occurring in July. The average precipitation is 1,068 mm, with rain falling mostly in summer. The reserve was listed as a Ramsar wetland in 2002 and joined the Asia-Australasia bird migration protection network in 2003 (*Jiang & Ding, 2011*). Due to the occurrence of intense anthropogenic economic activity, many areas along the Yellow Sea coastline have lost their original natural features (*Melville, Chen & Ma, 2016*). Our study site was located in the core area of the reserve, with almost no human disturbance, retaining the rare natural landscape of the Yellow Sea tidal flat.

## Vegetation distribution

A satellite image of the study site was first obtained from Google Earth Pro (Google LLC, Mountain View, CA, USA), followed by a landscape-scale analysis of the vegetation types (Fig. 1). Then, several plant characteristics that may affect the distribution of crabs were assessed during the peak of the growing season. These characteristics were measured in 10 randomly positioned one m² quadrats (1 × 1 m) in each of three habitat types (*P. australis*, *S. alterniflora* and *S. salsa*), and the distance between each quadrat was not less than 20 m. Within each quadrat, the number of culms was recorded, and 10 culms were randomly selected to measure the height using measuring tape. The number of culms per unit area represented the plant density.

## Crab sampling

To examine the distribution of crab communities in the tidal wetland, crabs were sampled from four marsh types (mudflat, *S. alterniflora*, *S. salsa*, *P. australis*). In our study area, the mean high tide level was approximately 4.68 m, and the mean low tide level was approximately 1.09 m. The sampled mudflat was approximately 1.5–2 m above sea level

and bare at low tide; the other three sampled habitats were approximately three to four m above sea level and flooded at high tide. From April to September 2017, pitfall traps were used to catch crabs once a month according to methods described in other recent studies (*Brusati & Grosholz, 2009*; *Cui, He & An, 2011*). We did not collect crabs during the winter because the crabs were not active due to the low temperatures. Four quadrats (20 × 20 m) were established in each of the four habitat types, and each quadrat in each habitat type was located more than 20 m away from the edge of the habitat. Six pitfall traps were deployed in each quadrat by burying cylindrical plastic buckets (20 cm in diameter, 30 cm deep) at the soil surface, and the distance between the buckets was not less than 10 m. After 1 week, the crabs in each trap were checked, identified to species, and counted. According to feeding classification studies describing crab functional groups (*Navarro-Barranco et al., 2013*; *Zhang et al., 2017*), the crabs were divided into three categories: carnivorous crabs, phytophagous crabs and omnivorous crabs. We did not obtain data in August because heavy rainfall destroyed the traps. Therefore, we only used the data from April, May, June, July and September for the final analysis.

## Data analysis

The differences in the characteristics of plants, the number of species and biomass of crabs and the Shannon diversity index of the crab communities among the four habitat types were examined using one-way ANOVA. The total biomass and Shannon diversity index values were calculated using the data from the crabs in each quadrat. To meet the assumptions of ANOVA, the numeric data were log or arcsine transformed prior to statistical analysis if necessary, and a Tukey HSD test followed if a significant difference was found. A principal component analysis (PCA) was performed to reduce the dimensions of the crab data to graphically analyze the patterns of the crab community structure according to habitat type. Each community was plotted as a point in space created by the first two PCA axes. Then, we tested for differences among habitats using one-way analysis of similarities (ANOSIM) (*Clarke & Warwick, 1994*) and used analysis of dissimilarity (SIMPER) to identify the species driving the dissimilarity among habitats (the relative contribution of individual species to the dissimilarity) (*Clarke & Warrick, 2001*). Before the ANOSIM and SIMPER analyses, the data were square root transformed to give equal weight to rare taxa and a Bray–Curtis similarity matrix was generated (*Bray & Curtis, 1957*). We also showed the temporal and spatial changes in the biomass of different crab species with multiple bar charts to evaluate the stability of the abundance of the individual crab species. All statistical analyses and graphing procedures were performed using the software R (version 3.5.1; *R Core Team, 2018*) with the FactoMineR (used for the PCA), vegan (used for the ANOSIM and SIMPER) and ggplot2 packages (*Wickham, 2016*).

## RESULTS

### Plant characteristics

In the different habitat types, there were significant differences in the plant height and stem density (Table 1). *P. australis* marsh had the greatest plant height ($F_{2, 297}$ = 805.4, $p < 0.001$), and *S. alterniflora* marsh had the greatest population density ($F_{2, 27}$ = 102.4,

**Table 1 Plant characteristics in three habitat types at Yancheng National Nature Reserve.**

| Plant characteristic | Habitat type | | | df | F | p |
|---|---|---|---|---|---|---|
| | Sp | Su | Ph | | | |
| Height (cm, $n = 100$) | $134.56 \pm 3.25^b$ | $31.94 \pm 0.81^c$ | $181.51 \pm 3.25^a$ | 2.297 | 805.4 | <0.001 |
| Density (stem·m$^{-2}$, $n = 10$) | $152.30 \pm 8.60^a$ | $19.3 \pm 1.95^c$ | $116.90 \pm 7.83^b$ | 2.27 | 102.4 | <0.001 |

Notes:
Shown are the mean ± SE. Different letters (a–c) indicate significant differences between habitat types ($p < 0.05$).
Habitat types: Sp, *S. alterniflora* marsh; Su, *S. glauca* marsh; Ph, *P. australis* marsh.

**Table 2 Density of crab species in four habitat types at Yancheng National Nature Reserve.**

| Crab species (num/per quadrat, $n = 20$) | Habitat type | | | | df | F | p |
|---|---|---|---|---|---|---|---|
| | Mu | Sp | Su | Ph | | | |
| 1. *Eriocheir sinensis* (C) | $1.95 \pm 0.37^b$ | $3.00 \pm 0.46^b$ | $2.05 \pm 0.31^b$ | $17.10 \pm 1.79^a$ | 3.76 | 60.22 | <0.001 |
| 2. *Chiromantes haematochir* (P) | — | $7.10 \pm 2.02$ | $8.60 \pm 1.31$ | $6.00 \pm 0.72$ | 2.57 | 0.812 | 0.449 |
| 3. *Chiromantes dehaani* (P) | — | $10.60 \pm 2.49$ | $6.40 \pm 1.58$ | $9.20 \pm 1.18$ | 2.57 | 1.36 | 0.265 |
| 4. *Helice tientsinensis* (P) | $2.05 \pm 0.37^c$ | $6.75 \pm 1.36^b$ | $11.65 \pm 1.68^a$ | $6.50 \pm 0.80^{b, c}$ | 3.76 | 11.28 | <0.001 |
| 5. *Metaplax longipes* (P) | $1.55 \pm 0.32$ | — | — | — | — | — | — |
| 6. *Sesarma plicata* (O) | — | $0.95 \pm 0.22^b$ | $3.5 \pm 0.49^a$ | — | 1.38 | 22.47 | <0.001 |
| 7. *Philyra pisum* (O) | $56.85 \pm 17.81$ | — | — | — | — | — | — |
| 8. *Uca arcuata* (P) | — | $19.65 \pm 1.95^a$ | $5.70 \pm 1.03^b$ | — | 1.38 | 39.91 | <0.001 |
| 9. *Cleistostoma dilatatum* (P) | — | — | $7.00 \pm 0.94$ | — | — | — | — |
| 10. *Macrophthalmus dilatatus* (P) | $30.50 \pm 7.26$ | — | — | — | — | — | — |

Notes:
Shown are the mean ± SE. Different letters (a–c) indicate significant differences between habitats ($p < 0.05$). "—" Indicates that no sample obtained.
Habitat types: Mu, Mudflat; Sp, *S. alterniflora* marsh; Su, *S. salsa* marsh; Ph, *P. australis* marsh. Functional feeding groups: C, Carnivorous group; P, Phytophagous group; O, Omnivorous group.

$p < 0.001$). The plant height and density in *S. salsa* marsh were significantly lower than those in the other two habitats.

## Composition of crab species

A total of 10 crab species were found among all habitats, with five species in the mudflat, six species in *S. alterniflora* marsh, seven species in *S. salsa* marsh and four species in *P. australis* marsh. Most species belonged to the phytophagous group, while the carnivorous group was represented by one species, and the omnivorous group contained two species (Table 2). There were significant differences in the composition of the crab communities and the density of each crab species in the different habitat types (Table 2). The dominant species in the mudflat were *Philyra pisum* and *Macrophthalmus dilatatus*, but there were no obvious dominant species in the other three habitat types. *P. pisum*, *M. dilatatus* and *Metaplax longipes* were only found in the mudflat, *Cleistostoma dilatatum* was only found in *S. salsa* marsh, and *Eriocheir sinensis* and *Helice tientsinensis* were widespread in all habitat types (Table 2). The total biomass per trap was the greatest in the mudflat ($F_{3, 76} = 16.11$, $p < 0.001$). However, the Shannon diversity index value was lowest for the mudflat, and *S. salsa* marsh had the highest biodiversity ($F_{3, 76} = 61.09$, $p < 0.001$) (Fig. 2).

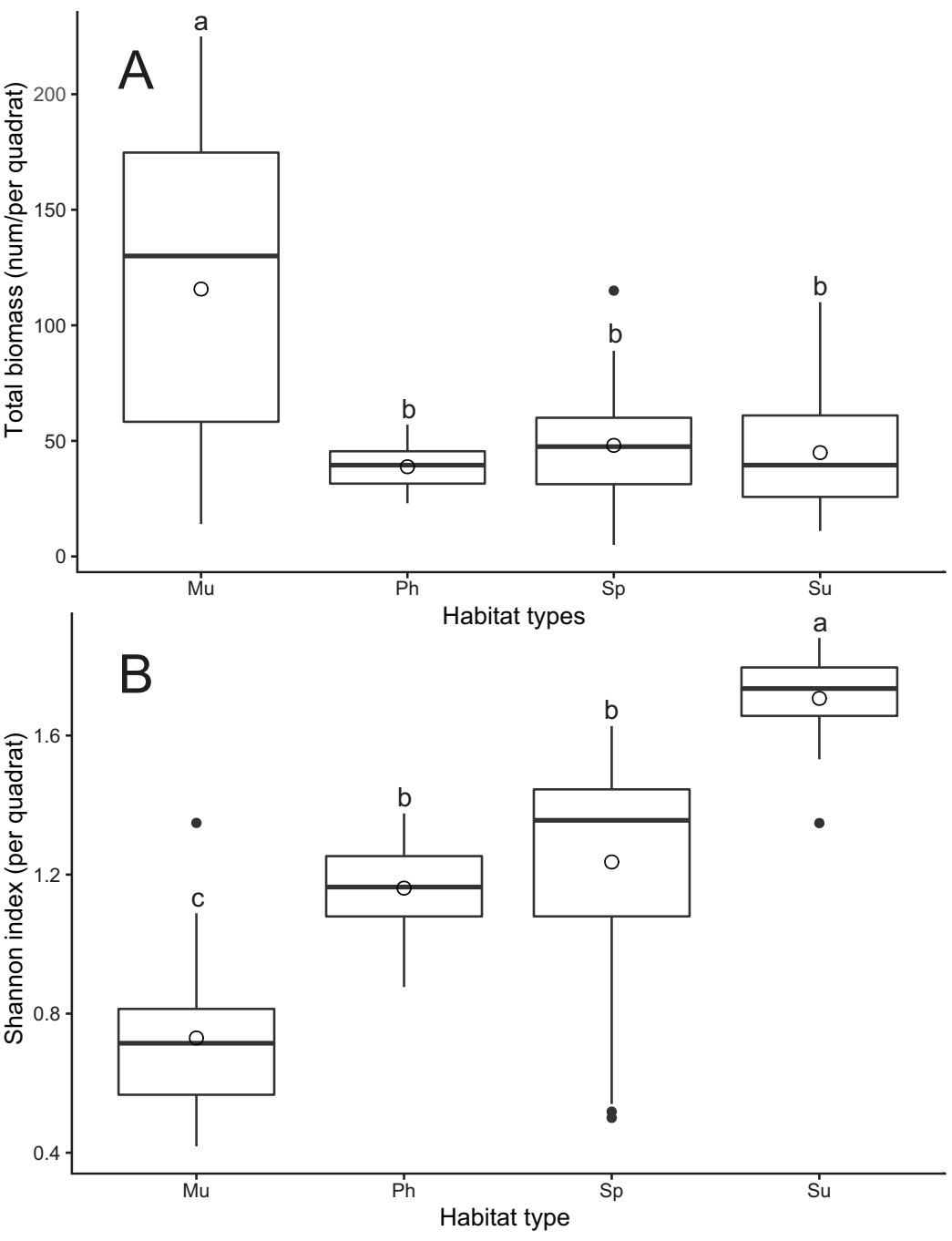

**Figure 2 Differences in crab biomass (A) and Shannon diversity index (B) among four habitat types at Yancheng National Nature Reserve.** Different letters (a–c) indicate significant differences between habitat types ($p < 0.05$). Habitat types: Mu, mudflat; Sp, *S. alterniflora* marsh; Su, *S. salsa* marsh; Ph, *P. australis* marsh.

### Crab species in *S. alterniflora* marsh

The crabs in *S. alterniflora* marsh included *E. sinensis*, *Chiromantes haematochir*, *Chiromantes dehaani*, *H. tientsinensis*, *Sesarma plicata* and *Uca arcuata* (Table 2). With the exception of *E. sinensis*, all other species belonged to the phytophagous group.

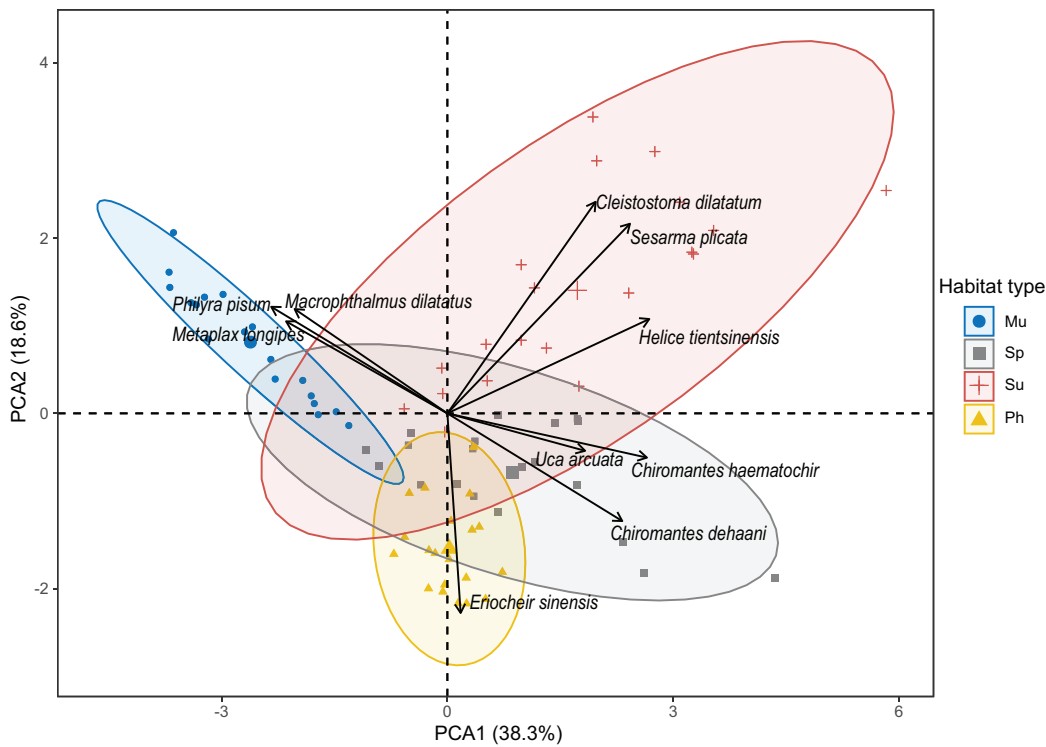

**Figure 3 Plot of the principal component analysis of crab community structure by the first two PCA axes.** Habitat types: Mu, mudflat; Sp, *S. alterniflora* marsh; Su, *S. salsa* marsh; Ph, *P. australis* marsh.

*E. sinensis* and *H. tientsinensis* were common in all four habitat types, the biomass of *E. sinensis* in *S. alterniflora* marsh was only significantly lower than that in *P. australis* marsh ($F_{3, 76} = 60.22$, $p < 0.001$), and the biomass of *H. tientsinensis* in *S. alterniflora* marsh was significantly higher than that in the mudflat but lower than that in *S. salsa* marsh ($F_{3, 76} = 11.28$, $p < 0.001$). *C. haematochir* and *C. dehaani* were found in all habitat types except the mudflat, and the biomass of these two species in *S. alterniflora* marsh was not significantly different from that in the other habitat types ($F_{2, 57} = 0.812$, $p = 0.449$; $F_{2, 57} = 1.36$, $p = 0.265$). *S. plicata* and *U. arcuata* were only found in *S. alterniflora* marsh and *S. salsa* marsh, the biomass of *S. plicata* in *S. alterniflora* marsh was lower than that in *S. salsa* marsh ($F_{1, 38} = 22.47$, $p < 0.001$), while the biomass of *U. arcuata* in *S. alterniflora* marsh was greater than that in *S. salsa* marsh ($F_{1, 38} = 39.91$, $p < 0.001$) (Table 2).

## Differences in crab community structure according to habitat type

According to the result of the PCA, the first two principal components accounted for a total of 56.9% of the crab community variation among habitats; PC1 explained 38.3% of the variation, and PC2 explained 18.6% of the variation (Fig. 3). The points representing the crab communities in *S. salsa* marsh were the most widely scattered and covered the largest area across the coordinate axis, while the points representing those in the mudflat were the most concentrated and covered the least area on the coordinate axis.

**Table 3 Results of ANOSIM and SIMPER for community structure by habitat type.**

| Group | ANOSIM | | SIMPER | | | | | | |
| | R | p | Dissimilarity (%) | Discriminating species 1 | Contribution (%) | Discriminating species 2 | Contribution (%) | Discriminating species 3 | Contribution (%) |
|---|---|---|---|---|---|---|---|---|---|
| Mu-Sp | 1 | 0.001 | 79.91 | *P. pisum* | 21.86 | *M. dilatatus* | 17.69 | *U. arcuata* | 16.93 |
| Mu-Su | 1 | 0.001 | 80.88 | *P. pisum* | 18.78 | *M. dilatatus* | 15.14 | *C. haematochir* | 11.12 |
| Mu-Ph | 1 | 0.001 | 74.56 | *P. pisum* | 24.97 | *M. dilatatus* | 20.22 | *C. dehaani* | 16.03 |
| Sp-Su | 0.55 | 0.001 | 30.80 | *C. dilatatum* | 28.46 | *C. dehaani* | 13.60 | *S. plicata* | 13.23 |
| Sp-Ph | 0.75 | 0.001 | 37.57 | *U. arcuata* | 37.85 | *E. sinensis* | 14.95 | *C. dehaani* | 13.15 |
| Su-Ph | 0.99 | 0.001 | 37.86 | *C. dilatatum* | 24.56 | *U. arcuata* | 21.81 | *S. plicata* | 19.82 |

Notes:
Global $R = 0.85$; $p = 0.001$; Permutation $N = 999$.
Habitat types: Mu, Mudflat; Sp, *S. alterniflora* marsh; Su, *S. salsa* marsh; Ph, *P. australis* marsh.

The area containing the points representing *S. alterniflora* marsh had the largest overlap with that of *S. salsa* marsh but had the lowest overlap with that of the mudflat (Fig. 3). The biomass of *S. plicata*, *H. tientsinensis* and *C. dilatatum* was positively loaded on PC1 and PC2, showing that these species preferred *S. salsa* marsh. The biomass of *C. haematochir*, *C. dehaani* and *U. arcuata* was positively loaded on PC1 but negatively loaded on PC2, showing that these species preferred *S. alterniflora* marsh. The biomass of *P. pisum*, *M. longipes* and *M. dilatatus* was positively loaded on PC2 but negatively loaded on PC1, showing that these species preferred the mudflat. The biomass of *E. sinensis* was negatively loaded on PC2 but was poorly positively loaded on PC1, showing that this species preferred *P. australis* marsh (Fig. 3).

The ANOSIM revealed significant differences in crab community structure among the habitat types (both globally and for pairwise tests) (global $R = 0.849$, $p < 0.001$) (Table 3). There were considerably significant differences between the mudflat and the three other habitat types in the pairwise tests (all $R = 1.0$, $p < 0.001$). The difference between *S. alterniflora* marsh and *S. salsa* marsh was the smallest, but it also reached a significant level ($R = 0.537$, $p < 0.001$) (Table 3).

The SIMPER revealed dissimilarity in the crab community structure ranging from 30.80% to 80.88% among habitat types; the dissimilarity between *S. alterniflora* marsh and *S. salsa* marsh was the lowest (30.8%) (Table 3). The species that were responsible for the dissimilarity between *S. alterniflora* marsh and *S. salsa* marsh were *C. dilatatum*, *C. dehaani* and *S. plicata* (representing 28.46%, 13.60% and 12.33%, respectively). The species that were responsible for the high dissimilarity (79.91%) between *S. alterniflora* marsh and the mudflat were *P. pisum*, *M. dilatatus* and *U. arcuata* (representing 21.86%, 17.69% and 16.93%, respectively). The species responsible for the dissimilarity (79.91%) between *S. alterniflora* marsh and *P. australis* marsh were *U. arcuata*, *E. sinensis* and *C. dehaani* (representing 37.85%, 14.95% and 13.15%, respectively) (Table 3).

## Temporal and spatial variation in individual species

The multiple bar charts showed that the compositions of the crab communities in each habitat type did not change in different sampling months, but the biomass of individual

species changed over time (Fig. 4). The biomass of crabs was most abundant from May to July in almost all habitats. *E. sinensis* represented the most biomass in May and July, and the biomass in *P. australis* marsh was always significantly higher than that in the three other habitats in all months (Fig. 4A). *C. haematochir* represented the most biomass in July, and the biomass in *S. alterniflora* marsh was the highest in July and September but was lower than that in other habitats in April, May and June (Fig. 4B). *C. dehaani* represented the most biomass in July, and the biomass in *S. alterniflora* marsh was the highest among all habitats in June and July (Fig. 4C). *H. tientsinensis* represented the most biomass in June and July, and the biomass in *S. alterniflora* marsh was the highest in June, while the biomass in *S. salsa* marsh was the highest in July among all habitats (Fig. 4D). *S. plicata* represented the most biomass in July, and the biomass in *S. alterniflora* marsh was always significantly lower than that in *S. salsa* marsh in all months (Fig. 4F). *U. arcuata* represented the most biomass in July, and the biomass in *S. alterniflora* marsh was always significantly higher than that in *S. salsa* marsh in all months (Fig. 4H). Of the three endemic species of the mudflat, *M. longipes* represented the most biomass in May and July, while *P. pisum* and *M. dilatatus* represented the most biomass in June and July (Figs. 4E, 4G and 4J). *C. dilatatum* was only found in *S. salsa* marsh and represented the most biomass in September (Fig. 4I).

## DISCUSSION

Although many studies have suggested that *S. alterniflora* invasion has resulted in local biodiversity loss and ecosystem degradation (*Li et al., 2009*; *Yang et al., 2017*), a unified conclusion regarding the effects of *S. alterniflora* invasion on macrofaunal species has not been reached (*Quan et al., 2016*). Recent studies have indicated that the reclamation of soil by *S. alterniflora* invasion can facilitate some taxa, such as fishes and crabs (*Feng et al., 2015*; *Chen & Ma, 2018*). Some studies confirm that the leaves of *S. alterniflora* can supply palatable food for *C. dehaani* and *S. plicata* (*Gao et al., 2018*; *Wang et al., 2008*) and that the abundance of *U. arcuata* is positively correlated with *S. alterniflora* cover (*Raposa et al., 2018*). According to our results, the number of crab species in *S. alterniflora* marsh was lower than that in *S. salsa* marsh but greater than that in the mudflat and *P. australis* marsh. The total biomass of crabs in *S. alterniflora* marsh was lower than that in the mudflat but higher than that in *P. australis* marsh and *S. salsa* marsh. The Shannon diversity index value of crabs in *S. alterniflora* marsh was higher than that for the mudflat and *P. australis* marsh and lower than that for *S. salsa* marsh. Therefore, we think that exotic *S. alterniflora* can offer suitable habitat for some crab species, especially those in the phytophagous group. For example, *C. haematochir*, *C. dehaani* and *U. arcuata* were more abundant in *S. alterniflora* marsh than in the other habitat types.

Despite the fact that *S. alterniflora* can create suitable environmental conditions that are consistent with the habitat requirements of some crab species (*Gao et al., 2018*; *Wang et al., 2008*), invasion may reduce the ecological niches of habitat-specific crabs. *S. alterniflora* is more competitive than native plants, especially in the mudflat and *S. salsa* marsh, and rapidly replaces these two habitat types (*Zuo et al., 2012*). According to

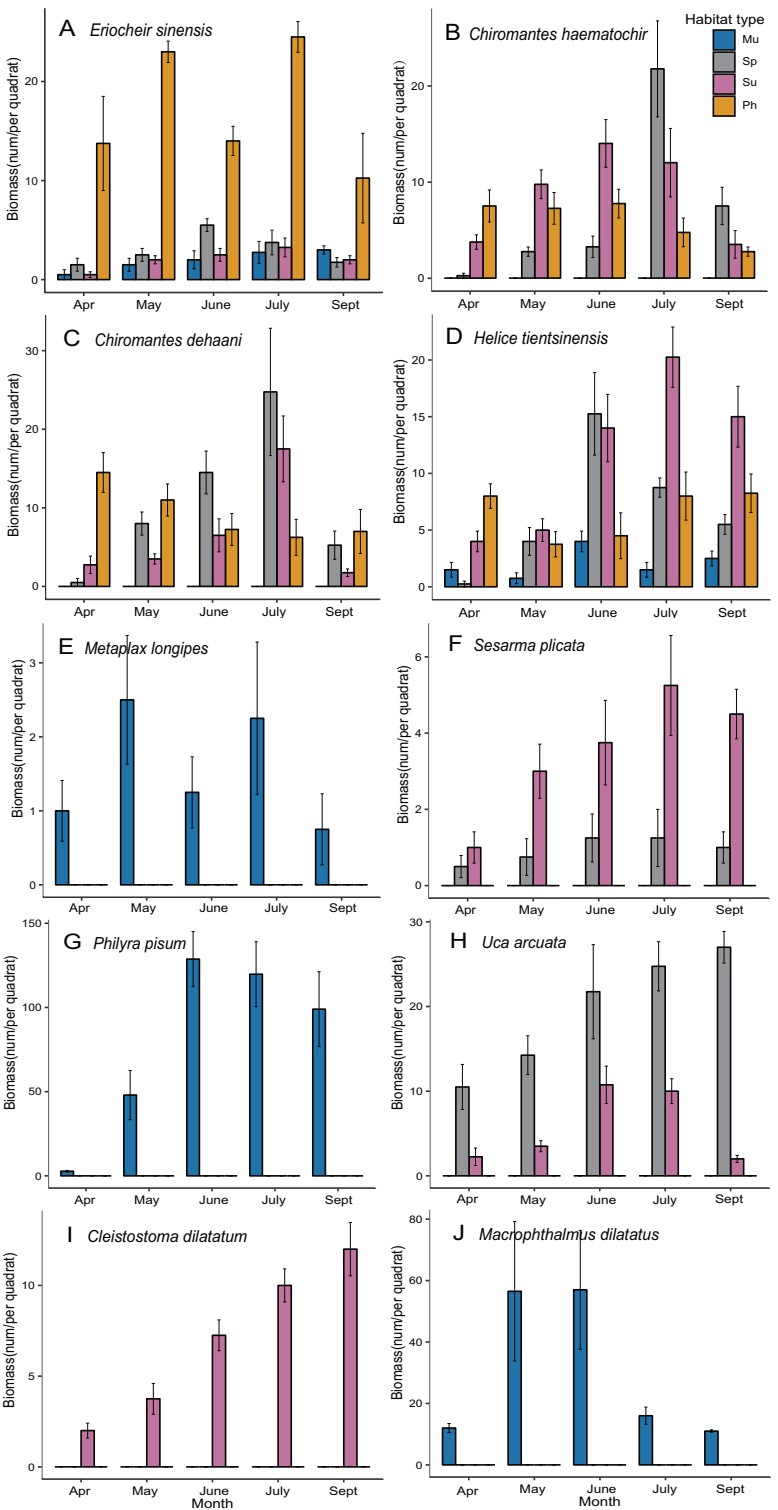

**Figure 4 Plots of spatiotemporal variation of individual species biomass.** (A–J) Biomass monthly dynamics of each crab species. Habitat types: Mu, mudflat; Sp, *S. alterniflora* marsh; Su, *S. salsa* marsh; Ph, *P. australis* marsh.

our results, the crab biomass in the mudflat was the largest among all habitat types (Fig. 2); for example, *M. longipes*, *P. pisum* and *M. dilatatus* only preferred the mudflat habitat (Table 2; Fig. 4). These species prefer to forage in low-altitude flooded habitats, but alien *S. alterniflora* accelerated sediment deposition and increased the elevation in invaded areas (*Chen, 2014*). The Shannon diversity index value for crabs was highest in *S. salsa* marsh (Fig. 2), and *C. dilatatum* was only found in *S. salsa* marsh (Table 2; Fig. 4), perhaps because *S. alterniflora* marsh does not provide the preferred food of this species. The loss of suitable habitat may cause the dramatic decline of populations of these crab species. The crab community structure in *S. alterniflora* marsh was significantly different from that in the mudflat according to the results of the PCA, ANOSIM and SIMPER (Fig. 3; Table 3), which means that even though *S. alterniflora* marsh hosts a considerable number of crabs, this habitat containing exotic vegetation may not perform the same ecological function as the mudflat.

Exotic plant invasions may have a cascading effect on higher trophic levels and ultimately impair overall biodiversity (*Brusati & Grosholz, 2009*). The effects of *S. alterniflora* invasion on the distribution of crabs may have a major impact on other related biological groups in the food web (*Gittman & Keller, 2013*). The wetlands along the Yellow Sea coast of China are a vital feeding habitat for shorebirds in the East Asian–Australasian Flyway (*Melville, Chen & Ma, 2016*). During the annual migration period, a large number of shorebirds rely on the predation of benthos to gain energy to complete migration (*Hou, Yu & Lu, 2013*). Many studies have confirmed that the highly dense structure of *S. alterniflora* is not suitable for medium and large shorebirds (*Delach, 2006*), and most native birds avoid using *S. alterniflora* marsh (*Ma et al., 2011*). Therefore, although *S. alterniflora* marsh is rich in crab resources according to our results, it seems to offer little help to migratory birds. Many plovers and dunlins prefer to forage on the mudflat (*Buchanan, 2003*; *Melville, Chen & Ma, 2016*), and some large, rare birds prey on crabs in *S. salsa* marsh (*Hemmi et al., 2006*), such as *Grus japonensis* foraging on *H. tientsinensis* (*Li et al., 2014*). *S. alterniflora* invasion may seriously damage the feeding habitat of these waterfowl and cause a decrease in some bird populations because of the inadequate food supply. In contrast, the adverse foraging environment of *S. alterniflora* marsh may provide crabs with a refuge from birds (*Nomann & Pennings, 1998*; *Sueiro, Bortolus & Schwindt, 2012*), resulting in the higher biomass and diversity of crabs.

## CONCLUSIONS

This study examined the distribution of crab communities along a habitat gradient from the shoreline to inland areas on the Yellow Sea coast in China, where the invasion of *S. alterniflora* has seriously encroached on the mudflat and native vegetation. Our results reveal that the crab community in *S. alterniflora* marsh is slightly different from that associated with the local vegetation but shows a large difference from that in the mudflat. This study indicates that *S. alterniflora* marsh can offer suitable habitat for some crab species, while some habitat-specific crab species reject this new habitat. The effects of the distribution of crabs after *S. alterniflora* invasion on the regional ecosystem will need further study in the future.

## ACKNOWLEDGEMENTS

We thank Bin Liu and patrols at Yancheng National Natural Reserve for assistance and support in the field sampling.

### Funding

This work was supported by the National Natural Science Foundation of China (No. 31670432). The funders had no role in study design, data collection and analysis, decision to publish, or preparation of the manuscript.

### Grant Disclosure

The following grant information was disclosed by the authors:
National Natural Science Foundation of China: 31670432.

### Competing Interests

The authors declare that they have no competing interests.

### Author Contributions

- Pan Chen conceived and designed the experiments, performed the experiments, analyzed the data, contributed reagents/materials/analysis tools, prepared figures and/or tables, authored or reviewed drafts of the paper, approved the final draft.
- Yan Zhang conceived and designed the experiments, performed the experiments, analyzed the data, contributed reagents/materials/analysis tools, prepared figures and/or tables, authored or reviewed drafts of the paper.
- Xiaojing Zhu performed the experiments.
- Changhu Lu conceived and designed the experiments, authored or reviewed drafts of the paper, approved the final draft.

### Field Study Permissions

The following information was supplied relating to field study approvals (i.e., approving body and any reference numbers):

Our study received verbal permission from the administrator of Yancheng National Natural Reserve. Name: Bin Liu; Authority: Chief of Scientific Research Management Section.

### Data Availability

The raw measurements are provided in the Supplemental File.

### Supplemental Information

Supplemental information for this article can be found online at http://dx.doi.org/10.7717/peerj.6775#supplemental-information.

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
