# Peer review of "Distribution of crabs along a habitat gradient on the Yellow Sea coast after Spartina alterniflora invasion"

_PeerJ, doi:10.7717/peerj.6775_

## Round 0.1 · original submission · Major Revisions

The reviewers have recommended changes to some figures and asked for some clarifications.
Please use “crabs” as the plural form of crab throughout the manuscript. Both reviewers pointed out the need for careful English editing when revising the manuscript. In addition, I have the following comments:

L 42 the current family name is Poaceae
L 50 Which taxa were studied / reported on by Liu et al 2012?
L 61 This should not be described as a “view’; instead describe the specific conclusion(s) based on data that can be found in the study by Cui et al 2011.
L 91 Please clarify methods: for clonal plants it is not generally possible to determine the number of plants in an area or to count 10 plants. Genotyping is required to identify individuals. If you refer to grasses, it is common to used the term “culm” to refer to a stem.
L98 It is important to provide data on elevation of each habitat (mean and range) in terms of a standard reference such as mean high tide. After looking at figure 1 I suspect that elevations are not equivalent. In that case, interpretation (Discussion) needs to also consider the hypothesis that differences in crabs across habitats could related to crab elevational preferences instead of or in addition to vegetation type / density.
L100 Can you comment on whether this sampling method is likely to capture all species of crabs in the area or whether there may be certain types of crabs that might not be captured using this method?
L104 Were the same positions used for pitfall traps throughout the study? I am wondering whether crab populations in the immediate vicinity of each trap might become depleted over time (depending on the efficiency of the trap).
L122-125 It is not correct to refer to population density for clonal individuals. Do you mean culm density or more generally, stem density?
L 140 correct spelling is “arcuata” It needs to be changed throughout.
L142-143 Instead of telling us they were significantly different, it is more informative to tell us if they were significantly higher or lower.
L153 What does “relatively” mean? Can you say there were no statistical differences? One objective stated in the introduction was to identify temporal patterns, so I think this aspect of the data should be given a bit more quantitative attention.
L162 I think “eroding” is not the correct word here. It invades the mudflat.

Table 1 I think units should be stems not individuals

Figure 2 Please remove the shaded grid structure from the background of this graph because it is distracting. Use black lines for X and Y axes.

Figures 3 and 4 - Both reviewers felt that these figures do not effectively convey patterns in the data. I also agree that these figures are difficult to interpret and not typical of what is presented in ecological studies. Reviewer 1 recommends a different way to visual differences while reviewer 2 recommends different analysis methods that are more commonly used in community ecology to identify differences among communities. Accordingly, I request that the authors reconsider their presentation of these figures with the aim of developing an intuitive presentation that conveys important features and trends in the data. Another standard method could be principal components analysis (PCA) or one of its variants whereby each community is plotted as a point in space created by two uncorrelated axes (the first two PCA axes).

Reviewer 1 ·

Basic reporting

The article would benefit from revision by an English speaker as there are a number of instances of strange word use and incorrect subject-verb agreements.
Literature references are sufficient.
Figures 3 and 4 are extremely challenging to interpret in their current form. In particular, navigating between the figure legend and the figures themselves to figure out what the letters mean in the figures requires too much effort. I encourage the authors to move away from these analytical frameworks and instead simply present the data in figure 3 as multiple panels of bar charts that show the average densities of each species in each of the habitat types (i.e. one small figure for each species, with column color referring to habitat type). I would also group the crabs into functional groups (e.g. predators versus herbivores/scavengers) so that at least your reader can get an idea if there are certain functional groups that seem to thrive in the invaded habitat

Experimental design

The research question is well defined and relevant. The implications of this study are significantly limited by the fact that the data are only derived from a single site, however. It would help if the authors at least better justified why this field site is perhaps representative of other marsh systems in the area.

While I appreciate the temporal sequence of the monitoring - showing crab densities on a monthly basis - the authors do not make it clear why they do not sample during the winter. I imagine this is because the crabs are not active due to lower temperature. However it would be helpful if this were stated. In addition, can the authors propose hypotheses related to the temporal dynamics of the crab communities - for instance do they expect the community composition to be more stable in one habitat type relative to another based on the stability of environmental conditions or food availability across habitats? The authors could calculate crab community stability metrics to evaluate such a hypothesis. In essence, I would like to see them better leverage the temporal data that they do have.

In addition, it needs to be clear in the methods and figure legends if the data being presented has been summed across all pitfall traps deployed over the entire monitoring period and thus what their units of measurement are.

Validity of the findings

As mentioned above, I found the third and fourth figures impossible to interpret and strongly encourage the authors to change their approach to presenting these data.

The discussion and conclusions are fine in my opinion.

Additional comments

I encourage the authors to reach out for help with editing to take care of some of the issues with English language and calculate community stability metrics to evaluate the stability of individual crab species and the entire community over time to better leverage their data set. And, justify if/how their study site is representative of other invaded wetlands in the region.

Reviewer 2 ·

Basic reporting

It's clear that the authors are not native English speakers. The paper has a number of grammatical errors and typos. These could easily be fixed, but I am not going to try and do it on a pdf file. A native speaker could fix many of these problems fairly quickly. The article as a whole is interesting and provides background on the effects of Spartina alterniflora on crab communities.

Experimental design

The design is straightforward, sampling crabs in different habitats in pitfall traps.

Validity of the findings

The data is ok. I am not familiar with the analyses cluster analyses used and it is not clear to me how figures 3 and 4 are interpreted. I would encourage the authors to explore a multivariate approach such as ANOSIM and SIMPER to more rigorously test their patterns.

Additional comments

This paper would be suitable for publication after additional data analyses and heavy editing on grammar.

---

## Round 0.2 · accepted · Accept

This is a model revision -- all issues raised by reviewers were addressed and the quality of writing is excellent. I only noticed a few minor formatting issues: in references, line 401 the species name should be in italics and three of the journal references used capitalization for all title words.

# Reviewer 2 ·

Basic reporting

grammatical errors fixed, article much improved

Experimental design

appropriate

Validity of the findings

the multivariate stats are a strength, the results section is nicely written and presented

Additional comments

The authors have done a good job addressing my concerns, the multivariate stats in particular, and the paper reads much better now. I do not think addition revisions are needed.